

# Prognostic utility of submental B/M-Mode ultrasonography for swallowing function assessment in post-stroke pharyngeal dysphagia: a preliminary study

Meng Huang[1] and Tong Wu[2]

[1] Ultrasound, Affiliated Hospital of Shaanxi University of Chinese Medicine, Xianyang, Shaanxi, China
[2] Ultrasound, Affiliated Brain Hospital of Nanjing Medical University, Nanjing, Jiangsu, China

## ABSTRACT

**Objective.** To evaluate the prognostic value of hyoid bone kinematic parameters measured via submental B/M-mode ultrasonography in post-stroke pharyngeal dysphagia, aiming to identify reliable predictors of swallowing recovery.

**Methods.** This study included 46 stroke patients with pharyngeal dysphagia, diagnosed via videofluoroscopic swallowing study (VFSS), and treated at the Department of Rehabilitation Medicine, Affiliated Brain Hospital of Nanjing Medical University (June 2020–August 2024). Swallowing function was assessed using seven pharyngeal phase parameters from the Videofluoroscopic Dysphagia Scale (VDS) at baseline and post-rehabilitation. Patients were stratified into favorable ($n = 26$) and unfavorable ($n = 20$) prognosis groups based on post-treatment VDS and Functional Oral Intake Scale (FOIS) scores. Submental B/M-mode ultrasonography quantified hyoid bone displacement parameters pre- and post-rehabilitation, including maximum displacement, anterior displacement, superior displacement, total movement duration, and hyoid-thyroid cartilage approximation ratio (ASR).

**Result.** No significant differences were observed between groups in baseline characteristics. Post-treatment, the favorable group showed significantly lower Rosenbek scores ($p = 0.000$), reduced VDS scores ($p = 0.000$), and decreased feeding tube dependency ($p = 0.000$). Post-treatment, the favorable group exhibited greater anterior displacement ($p = 0.011$), higher ASR ($p = 0.000$), and shorter total movement duration ($p = 0.005$). Logistic regression identified hyoid anterior displacement (odds ratio (OR) = 9.539, $p = 0.011$) and ASR (OR = 14.238, $p = 0.001$) as independent prognostic predictors. ROC curve analysis indicated that hyoid anterior displacement (area under the curve (AUC) = 0.720) and ASR (AUC = 0.816) were significant discriminators of favorable outcomes, with optimal cutoff values of 0.865 cm (92.3% sensitivity, 50.0% specificity) and 31.5% (84.6% sensitivity, 65.0% specificity), respectively. The combined model further improved predictive accuracy (AUC = 0.854, 84.6% sensitivity, 85.0% specificity).

**Conclusion.** Impaired hyoid anterior displacement and reduced ASR are critical pathophysiological mechanisms in post-stroke dysphagia. Combined assessment of these parameters provides significant clinical utility for prognosis and treatment planning.

Corresponding author
Tong Wu, wutong19812013@163.com

## INTRODUCTION

Post-stroke dysphagia represents a prevalent and clinically significant complication, frequently leading to severe sequelae including aspiration pneumonia, nutritional deficiencies, and recurrent respiratory infections. These complications collectively contribute to increased morbidity and mortality rates, significantly impairing patients' quality of life and imposing substantial burden on healthcare systems (*Jones, Colletti & Ding, 2020*). The anterior displacement of the hyoid bone plays a crucial biomechanical role in swallowing physiology by facilitating cricopharyngeal opening through traction on the cricoid cartilage, thereby enabling efficient bolus transit through the upper esophageal sphincter (*Chen et al., 2023*).

Videofluoroscopic swallowing study (VFSS), widely regarded as the diagnostic gold standard for dysphagia assessment (*Martino, Pron & Diamant, 2000*), provides dynamic visualization of pharyngeal anatomy and bolus dynamics during deglutition. This imaging modality not only enables precise detection of aspiration events but also offers valuable insights into the pathophysiological mechanisms underlying swallowing dysfunction (*Tang et al., 2022*). Despite its diagnostic superiority, VFSS presents several clinical limitations, including ionizing radiation exposure and potential risks associated with contrast agent aspiration, which may lead to contrast-induced pulmonary complications in vulnerable populations (*Jo et al., 2015*).

Ultrasonography has emerged as a valuable diagnostic tool, offering distinct advantages including clinical practicality, portability, non-invasiveness, radiation-free operation, and the capacity to utilize regular food textures during assessment (*Bragato, Silva & Berti, 2024*). The kinematic analysis of hyoid bone movement represents a critical parameter in swallowing biomechanics evaluation, with ultrasound demonstrating significant potential as both a screening tool and quantitative assessment method for swallowing function, enabling precise measurement of swallowing duration and comprehensive characterization of hyoid bone displacement patterns (*Hsiao, Wahyuni & Wang, 2013*). This study aims to investigate the prognostic value of hyoid bone kinematic parameters measured by submental B/M-mode ultrasonography in patients with post-stroke pharyngeal dysphagia, with the ultimate objective of establishing reliable predictive indicators for clinical outcomes and therapeutic decision-making.

## METHODS

### Ethics and recruitment

The study received ethical approval from the Medical Ethics Committee of the Affiliated Brain Hospital of Nanjing Medical University (2020-KY162-01). Written informed consent was obtained from all participants prior to their inclusion in the study. A retrospective analysis was conducted on 57 patients with post-stroke dysphagia who met the inclusion criteria and exclusion criteria between June 2020 and August 2024 in the Department of

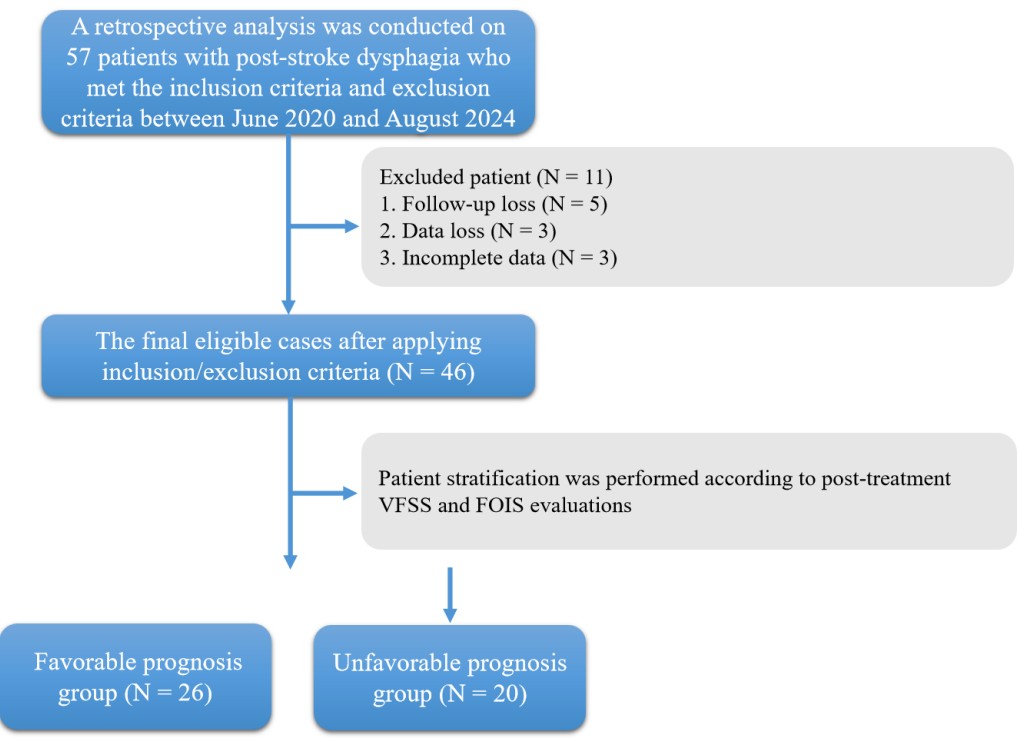

**Figure 1** Flowchart of patient enrollment and grouping for pharyngeal dysphagia after stroke.

Rehabilitation Medicine at the Affiliated Brain Hospital of Nanjing Medical University. Due to loss of follow-up, data loss or incomplete data, only 46 cases were left (Fig. 1). The diagnosis was confirmed through videofluoroscopic swallowing study (VFSS), which is considered the gold standard for dysphagia assessment.

## Inclusion and exclusion criteria

The inclusion criteria specified that patients must meet the diagnostic criteria for cerebrovascular diseases as outlined in the 2019 Chinese Guidelines for the Diagnosis and Treatment of Cerebrovascular Diseases, with cerebral infarction or hemorrhage confirmed by neuroimaging through MRI or CT (*The Neurology Branch of the Chinese Medical Association & the Cerebrovascular Disease Group of the Neurology Branch of the Chinese Medical Association, 2019*). Additionally, a diagnosis of pharyngeal dysphagia needed to be established according to the Chinese Rehabilitation Management Guidelines for Swallowing Disorders (2023 edition) (*Swallowing Disorder Rehabilitation Professional Committee of Chinese Association of Rehabilitation Medicine, 2023*), with confirmation *via* videofluoroscopic swallowing study (VFSS), which serves as the gold standard for dysphagia assessment.

The exclusion criteria encompassed several patient groups to ensure study validity. Patients with pre-existing conditions that could compromise swallowing function, such as Parkinson's disease, neurodegenerative disorders, brain tumors, or head and neck

malignancies, were excluded. Additionally, those presenting with severe systemic conditions or hemodynamic instability that contraindicated diagnostic procedures were not eligible. Further exclusions applied to patients with impaired consciousness, defined as a Glasgow Coma Scale score below 15, or significant cognitive impairment, indicated by a Mini-Mental State Examination score below 24, as these factors could affect examination compliance. Lastly, individuals with a history of cervical spine surgery or any surgical intervention involving the pharyngeal region were also excluded from participation.

## Participant demographics

The study cohort comprised 36 male and 10 female participants,with ages ranging from 47 to 83 years (mean age: $65.2 \pm 9.2$ years). The post-stroke duration at the time of enrollment varied from 2 weeks to 4.5 months, representing the subacute to chronic phases of stroke recovery.

## Clinical intervention methods

All participants initiated enteral nutrition support or dietary modifications based on VFSS findings and commenced comprehensive swallowing rehabilitation therapy immediately following initial VFSS assessment. The rehabilitation protocol, supervised by certified swallowing therapists for 30-minute daily sessions. The rehabilitation protocol, supervised by certified swallowing therapists for 30-minute daily sessions, consisted of: (1) conventional swallowing maneuvers including the Mendelsohn maneuver, Shaker exercise, supraglottic swallow technique, and Masako maneuver; (2) pharyngeal thermal-tactile stimulation combined with neuromuscular electrical stimulation (NMES); and (3) additional transnasal balloon dilation therapy for patients presenting with partial cricopharyngeal achalasia. The therapeutic regimen was administered once daily, five times per week, continuing until either patient discharge or complete restoration of swallowing function.

## Grouping

Patient stratification was performed according to post-treatment videofluoroscopic swallowing study (VFSS) and Functional Oral Intake Scale (FOIS) evaluations, utilizing specific classification criteria. Patient stratification was performed according to post-treatment VFSS and FOIS evaluations, utilizing the following classification criteria: (1) Favorable prognosis group: transition from initial VFSS-guided enteral feeding to oral intake following rehabilitation therapy, with VDS score <20 and FOIS score $\geq$ 5; (2) unfavorable prognosis group: persistent requirement for enteral feeding or intermittent tube feeding based on both initial and post-rehabilitation VFSS assessments, with VDS score $\geq$ 20 and FOIS score < 5 (*McMicken, Muzzy & Calahan, 2010*; *Kim et al., 2012*).

## Radiological examination methods

All enrolled patients were required to complete both videofluoroscopic swallowing study (VFSS) and ultrasonographic swallowing function assessment within 72 h post-admission, with comprehensive documentation of enteral feeding status and relevant clinical parameters.

## VFSS examination
### Equipment & positioning

The examination was performed using the Philips Azurion 7 B20 fluoroscopic system. The video perspective imaging function is activated. The specific parameters are kV/mA: 69.7/8.0, 30–60 f/s, and the contrast and brightness should be adjusted appropriately to clearly display the structure of the mouth, pharynx, esophagus and the flow of the contrast agent during swallowing. Patients were positioned in either an upright seated position or wheelchair-supported posture.

### Configuration of the test boluses

Under fluoroscopic guidance, the patient was instructed to swallow 1:1 (100 ml normal saline + 100 ml iodine contrast agent), medium thick (2.0%), low thick (1%) and high thick (3.0%) liquids made from iodine contrast agent (Omnipaque) and Super S thickener (NUTRI Corporation, Japan, xanthan gum-based) and the original solution of iohexol (1:1).

### Measured parameters

VFSS interpretation was conducted independently by two rehabilitation medicine associate chief physicians, with consensus reached through joint analysis in cases of discrepancy. The Video Fluoroscopic Dysphagia Scale (VDS) (*Kim et al., 2012*), which evaluates seven pharyngeal phase parameters (total score: 60; inverse correlation with swallowing function). The Rosenbek Penetration-Aspiration Scale (PAS) (*Smith et al., 1999*), with the following classification: grade 1 (no penetration/aspiration); grades 2–4 (penetration only); grades 5–8 (aspiration). For statistical analysis, numerical scores (1–8) were assigned, demonstrating positive correlation with aspiration severity. The Functional Oral Intake Scale (FOIS) (*Crary, Mann & Groher, 2005*) assesses feeding methods and nasogastric tube dependency (score range: 1–7; positive correlation with swallowing function).

## Ultrasonographic methodology
### Probe placement & positioning

Ultrasonographic assessments were performed using the Philips EPIQ 7 color Doppler ultrasound system equipped with a low-frequency curvilinear transducer (2–5 MHz). Image acquisition was standardized at a frame rate of 22.5 frames per second. Participants were positioned in an upright seated posture with firm back support, maintaining a neutral head position (Frankfort horizontal plane) and forward gaze. The transducer was carefully placed along the midsagittal plane of the submental region, with meticulous attention to minimizing probe pressure that could potentially influence swallowing kinematics.

### Recording & analysis

Patients were instructed to swallow three times while keeping their heads stable, with intervals of 1 to 2 min between each swallow. The video recording function was activated to capture the movement of the hyoid bone from its resting position to the point of maximum displacement and back to the original resting position. These videos were stored in AVI format. Using Kinovea software (for video microanalysis), tracer markers were placed on the hyoid bone to automatically track its displacement trajectory.

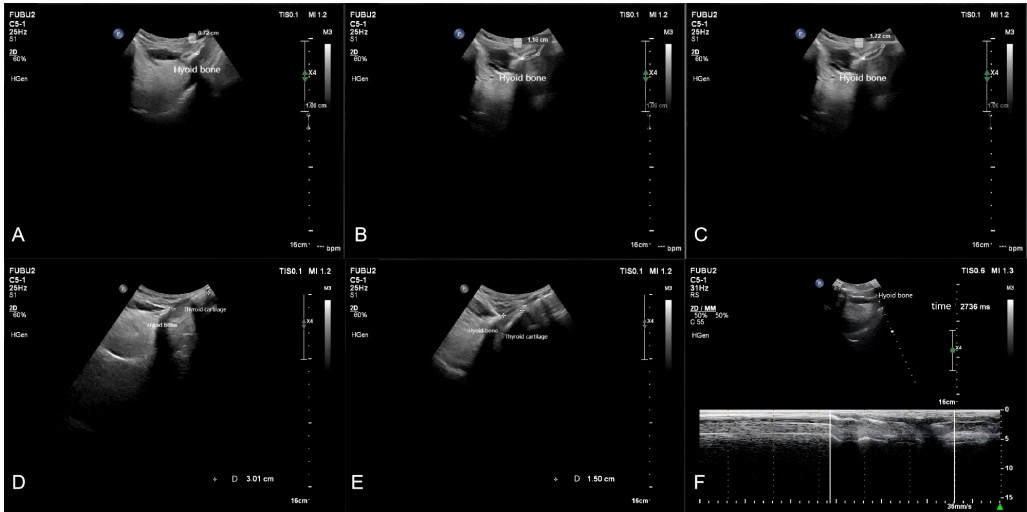

**Figure 2  Schematic representation of hyoid bone kinematics during swallowing in the favorable prognosis group under ultrasonographic assessment.**

### *Measured parameters*

Measurements were taken for the upward displacement, forward displacement, and maximum displacement of the hyoid bone. The low-frequency probe initiated M-mode ultrasonography to measure the duration of hyoid bone movement during swallowing. The maximum distance between the inferior border of the hyoid bone and the inferior border of the thyroid cartilage, as well as the minimum distance between them after swallowing, were measured. The hyoid-thyroid cartilage approximation ratio (ASR) was then calculated (Figs. 2 and 3). Based on the hyoid bone's movement trajectory, a two-dimensional coordinate graph was constructed (Fig. 4).

## Statistical analysis

Statistical analyses were performed using SPSS software (version 27.0; IBM Corp). The normality of data distribution was assessed using the Shapiro–Wilk test, while homogeneity of variance was evaluated through Levene's test. Normally distributed continuous variables were expressed as mean $\pm$ standard deviation ($\bar{x} \pm s$) and compared using independent samples $t$-test. Non-normally distributed continuous variables were presented as median (interquartile range) [M (Q1, Q3)] and analyzed using the Mann–Whitney U test. Categorical variables were expressed as frequencies and percentages, with between-group comparisons performed using chi-square test. Binary logistic regression analysis was conducted using the Enter method to identify significant predictors among ultrasonographic hyoid bone kinematic parameters in post-stroke pharyngeal dysphagia. The diagnostic performance of significant parameters was evaluated using receiver operating characteristic (ROC) curve analysis, with the area under the curve (AUC) calculated to determine predictive accuracy. All statistical tests were two-tailed, with $p$-values <0.05 considered statistically significant.

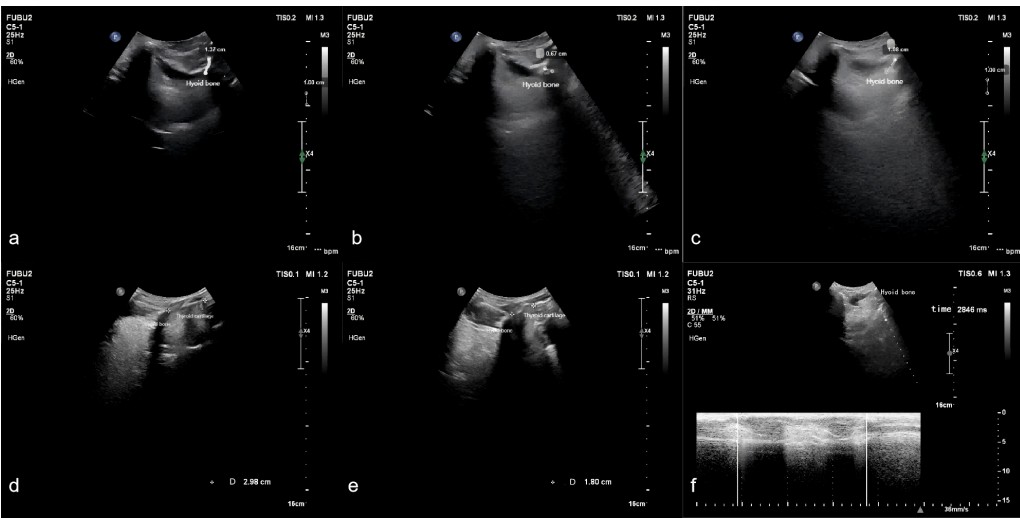

**Figure 3** Schematic representation of hyoid bone kinematics during swallowing in the unfavorable prognosis group under ultrasonographic assessment.

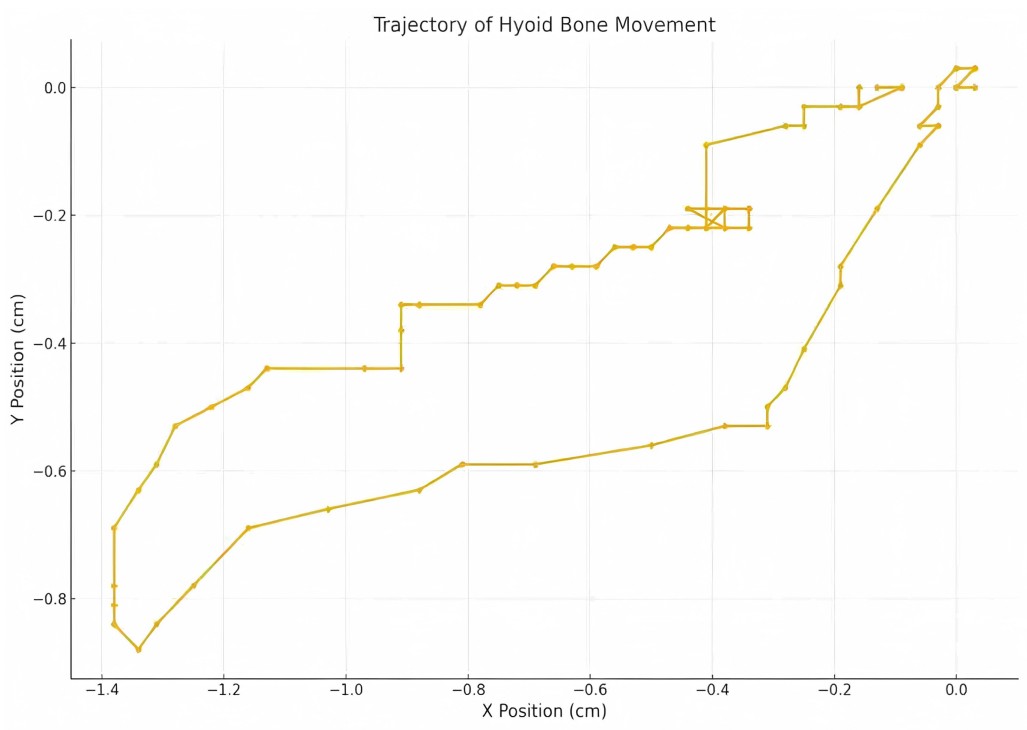

**Figure 4** Two dimensional coordinate system representation of hyoid bone displacement trajectory during deglutition.

**Table 1 Demographic information and clinical assessment of in the favorable prognosis group and the unfavorable prognosis group.**

| Group | Age (y) | Disease course (d) | Treatment duration (d) | Gender | | Stroke types | | Stroke location | |
|---|---|---|---|---|---|---|---|---|---|
| | | | | M | F | Infarction | Hemor-rhage | Above the tentorium cerebelli | Under the tentorium cerebelli |
| Favorable prognosis $n = 26$ | $61.1 \pm 15.3$ | 29.2 (23.7,98.5) | $48.4 \pm 46.1$ | 21 | 5 | 23 | 3 | 8 | 18 |
| Unfavorable prognosis $n = 20$ | $66.9 \pm 10.2$ | 76.6 (32.9,206.2) | $58.3 \pm 22.4$ | 15 | 5 | 15 | 5 | 5 | 15 |
| t/$\chi^2$ | −1.440 | −1.629 | −0.884 | 0.211 | | — | | 0.186 | |
| P | 0.157 | 0.103 | 0.343 | 0.638 | | 0.267 | | 0.667 | |

# RESULTS

## Demographics and clinical assessments at baseline

Table 1 summarizes the comparative analysis of demographic and clinical characteristics between the favorable prognosis group and unfavorable prognosis group, including gender distribution, mean age, stroke type (cerebral infarction *vs.* cerebral hemorrhage), lesion location, disease duration, and treatment period. No statistically significant differences were observed between the two groups in any of these parameters (Mann–Whitney U test, chi-square test, all $p > 0.05$).

## Comparison of FOIS scores, PAS scores, VDS scores, and enteral feeding status between the favorable and unfavorable prognosis groups at baseline and post-treatment time points

As presented in Table 2, baseline assessments revealed no statistically significant intergroup differences in FOIS scores, PAS scores, VDS scores, or enteral feeding dependency status between the favorable and unfavorable prognosis groups (independent samples *t*-test, chi-square test, *all p > 0.05*). However, post-treatment analysis showed significantly improved swallowing function, with lower PAS and VDS scores, and reduced enteral feeding dependency compared to the unfavorable prognosis group *(all p < 0.05)*.

## Comparison of ultrasonic hyoid kinematics parameters between the favorable and unfavorable prognosis groups at baseline and post-treatment time points

As presented in Table 3, baseline assessments revealed no statistically significant intergroup differences in hyoid bone kinematic parameters, including maximum displacement amplitude, anterior displacement amplitude, superior displacement amplitude, total movement duration, and ASR between the favorable and unfavorable prognosis groups (Mann–Whitney U test, *all p > 0.05*). Table 4 demonstrates that while post-treatment maximum displacement amplitude and superior displacement amplitude remained comparable between groups ($p > 0.05$), the favorable prognosis group exhibited significantly greater anterior displacement amplitude ($p = 0.011$), reduced total movement

**Table 2  Comparison of FOIS scores, Rosenbek scores, VDS scores, and enteral feeding status between the favorable and unfavorable prognosis groups at baseline and post-treatment time points.**

| Group | FOIS | | PAS | | VDS | | Enteral feeding (with/without) | |
|---|---|---|---|---|---|---|---|---|
| | Baseline | Post-treatment | Baseline | Post-treatment | Baseline | Post-treatment | Baseline | Post-treatment |
| Favorable prognosis $n = 26$ | $2.58 \pm 1.27$ | $6.08 \pm 0.84$ | $6.38 \pm 1.13$ | $3.50 \pm 1.50$ | $42.35 \pm 6.94$ | $19.04 \pm 10.60$ | 14/12 | 2/24 |
| Unfavorable prognosis $n = 20$ | $1.85 \pm 1.14$ | $2.90 \pm 1.02$ | $6.90 \pm 1.02$ | $6.50 \pm 1.05$ | $45.25 \pm 4.36$ | $42.60 \pm 3.95$ | 16/4 | 18/12 |
| $Z/\chi^2$ | $-1.923$ | $-5.758$ | $-1.582$ | $-5.071$ | $-1.123$ | $-5.367$ | — | — |
| $P$ | 0.054 | 0.000 | 0.114 | 0.000 | 0.262 | 0.000 | 0.117 | 0.000 |

**Table 3  Ultrasonic hyoid kinematics parameters in the favorable prognosis group and the unfavorable prognosis group at baseline.**

| Group | Hyoid bone maximum displacement (cm) | Hyoid bone anterior displacement (cm) | Hyoid bone superior displacement (cm) | Hyoid bone movement duration (s) | ASR(%) |
|---|---|---|---|---|---|
| Favorable prognosis $n = 26$ | 1.34 (1.20, 1.52) | 0.97 (0.79, 1.26) | 0.92 (0.71, 1.13) | 1.18 (0.94, 1.61) | 0.31 (0.19, 0.37) |
| Unfavorable prognosis $n = 20$ | 1.38 (1.06, 1.53) | 0.83 (0.66, 1.18) | 1.09 (0.63, 1.29) | 1.24 (1.02, 2.83) | 0.29 (0.23, 0.38) |
| $Z$ | $-1.826$ | $-0.044$ | $-1.031$ | $-0.177$ | $-1.097$ |
| $P$ | 0.068 | 0.965 | 0.303 | 0.859 | 0.273 |

duration ($p = 0.005$), and higher ASR values ($p = 0.000$) compared to the unfavorable prognosis group.

## Impact of ultrasonographic hyoid bone kinematic parameters on prognosis in post-stroke pharyngeal dysphagia

Binary logistic regression analysis of the ultrasonographic hyoid bone kinematic parameters identified anterior displacement amplitude (odds ratio (OR) = 9.539, $p = 0.011$) and ASR (OR = 14.238, $p = 0.001$) as significant independent predictors of swallowing outcomes in post-stroke pharyngeal dysphagia. The complete regression analysis results are presented in Table 5.

## Predictive significance of ultrasonic hyoid kinematic parameters on the prognosis of patients with post-stroke pharyngeal swallowing disorders

Receiver operating characteristic (ROC) curve analysis revealed that hyoid anterior displacement demonstrated moderate predictive accuracy (AUC = 0.720) for post-stroke pharyngeal dysphagia prognosis, with optimal diagnostic threshold at 0.865 cm (sensitivity: 92.3%, specificity: 50.0%). ASR showed good predictive performance (AUC

**Table 4** Ultrasonic hyoid kinematics parameters in the favorable prognosis group and the unfavorable prognosis group at post-treatment time points.

| Group | Hyoid bone maximum displacement (cm) | Hyoid bone anterior displacement (cm) | Hyoid bone superior displacement (cm) | Hyoid bone movement duration (s) | ASR (%) |
|---|---|---|---|---|---|
| Favorable prognosis $n = 26$ | 1.65 (1.49, 2.07) | 1.39 (1.17, 1.54) | 1.21 (0.84, 1.42) | 0.95 (0.88, 1.18) | 0.41 (0.32, 0.46) |
| Unfavorable prognosis $n = 20$ | 1.59 (1.20, 1.89) | 0.94 (0.67, 1.42) | 1.14 (0.75, 1.37) | 1.76 (1.00, 2.76) | 0.28 (0.24, 0.34) |
| Z | −1.219 | −2.538 | −0.455 | −2.814 | −3.649 |
| P | 0.223 | 0.011 | 0.649 | 0.005 | 0.000 |

**Table 5** LOGISTIC regression analysis of ultrasound hyoid kinematic parameters.

| Independent variable | B | SE | Wald | P value | OR | 95% CI |
|---|---|---|---|---|---|---|
| Hyoid bone maximum displacement (cm) | 1.238 | 0.759 | 2.662 | 0.103 | 3.449 | 0.779~15.261 |
| Hyoid bone anterior displacement (cm) | 2.255 | 0.889 | 6.439 | 0.011 | 9.539 | 1.671~54.458 |
| Hyoid bone superior displacement (cm) | 0.494 | 0.816 | 0.367 | 0.545 | 1.639 | 0.331~8.110 |
| Hyoid bone movement duration (s) | 2.118 | 0.878 | 6.205 | 0.103 | 1.112 | 0.020~1.627 |
| ASR (%) | 2.565 | 0.772 | 11.824 | 0.001 | 14.238 | 3.133~64.702 |

**Notes.**
In the table, B represents the regression coefficient, SE represents the standard error, Wald represents the Wald chi-square value, OR represents the odds ratio for exposure to risk factors, and 95% CI represents the 95% confidence interval.

**Table 6** Predictive value of ultrasonic hyoid kinematic parameters for the prognosis of patients with post-stroke pharyngeal dysphagia.

| | AUC | Yorden index | Sensitivity (%) | Specificity (%) | P value | 95% CI |
|---|---|---|---|---|---|---|
| Hyoid bone anterior displacement (cm) | 0.720 | 0.423 | 92.3 | 50.0 | 0.011 | 0.564~0.876 |
| ASR (%) | 0.816 | 0.496 | 84.6 | 65.0 | 0.005 | 0.095~0.416 |
| The combination of two parameters | 0.854 | 0.696 | 84.6 | 85.0 | 0.000 | 0.730~0.978 |

= 0.816) with an optimal cutoff value of 31.5% (sensitivity: 84.6%, specificity: 65.0%). The combined model incorporating both parameters exhibited superior predictive value (AUC = 0.854), achieving balanced sensitivity (84.6%) and specificity (85.0%). Complete ROC analysis results are presented in Table 6. The integrated predictive efficacy of anterior displacement amplitude and ASR for post-stroke pharyngeal dysphagia prognosis significantly outperformed individual parameter assessments (Fig. 5).

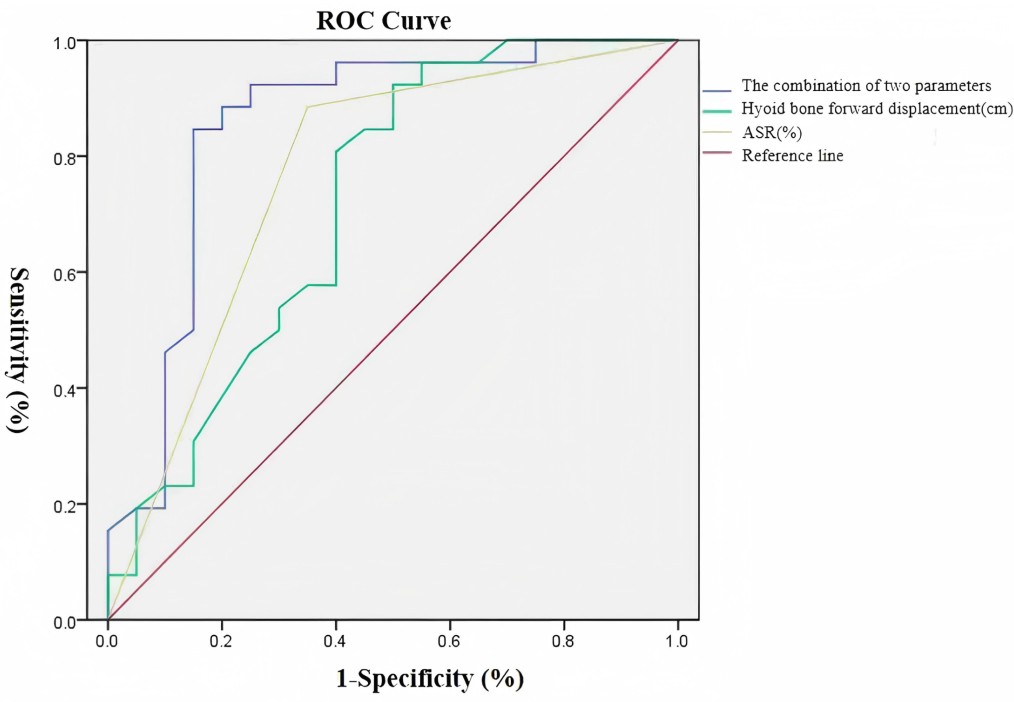

**Figure 5** Receiver operating characteristic (ROC) curves of ultrasonographic hyoid bone kinematic parameters for prognostic prediction in post-stroke pharyngeal dysphagia.

## DISCUSSION

Swallowing represents a sophisticated and rapid neuromuscular process that requires precise coordination among multiple structures within the oral cavity, pharynx, and esophagus (*Mistry & Hamdy, 2008*). This complex mechanism is typically categorized into three distinct phases: oral, pharyngeal, and esophageal. Notably, dysfunctions occurring during the pharyngeal phase may result in critical clinical complications, including aspiration and airway obstruction (*Paik et al., 2008*). The hyoid bone, mandible, and tongue form a functionally integrated complex that plays a pivotal role in both mastication and deglutition processes (*Auvenshine & Pettit, 2020*). The pharyngeal phase is initiated by the contraction of the suprahyoid muscle group, which generates simultaneous anterior-superior displacement of the hyoid bone. This movement facilitates two critical protective mechanisms: epiglottic inversion and laryngeal vestibule closure.

The spatial relationship between the hyoid bone and thyroid cartilage, as measured by the hyoid-thyroid distance, represents a crucial parameter for assessing laryngeal elevation during swallowing (*Kuh et al., 2003*). In recent years, ultrasonography has emerged as a valuable diagnostic tool for assessing swallowing disorders, offering distinct advantages including cost-effectiveness, non-invasiveness, radiation-free operation, and excellent reproducibility (*Kwak et al., 2018a*). This imaging modality enables precise visualization of hyoid bone kinematics during swallowing. Comparative studies have established a strong concordance between ultrasound-derived hyoid bone movement trajectories and those

obtained through VFSS (*Yabunaka et al., 2011*). Quantitative ultrasound measurements of hyoid displacement have demonstrated robust criterion validity, showing high correlation coefficients with VFSS measurements, along with excellent intra-rater and inter-rater reliability indices (*Kwak et al., 2018b*). Furthermore, ASR as a normalized parameter, demonstrates reduced susceptibility to individual anatomical variations.

Previous investigations have elucidated a significant association between hyoid bone displacement dynamics and the incidence of penetration-aspiration during deglutition (*Zhang et al., 2020*). This is due to the fixed sequence of contractions in the muscles adjacent to the hyoid during the pharyngeal phase of swallowing. Any damage to the swallowing-related nervous system controlling these muscles can disrupt this sequence. Reduced anterior movement of the hyoid, caused by impairment of the swallowing-related nervous system, may lead to poor relaxation of the upper esophageal sphincter, accumulation of residue in the pyriform sinus, and increased risk of aspiration (*Lee et al., 2021*).

The results of this study indicate that in the unfavorable prognosis group after treatment, both the anterior displacement of the hyoid ($p = 0.011$) and the ASR ($p = 0.000$) were significantly lower than those in the favorable prognosis group. Moreover, the movement time of the hyoid was significantly longer in the unfavorable prognosis group ($p = 0.005$). Logistic regression analysis combined with ROC curve analysis revealed that anterior displacement amplitude of the hyoid and the hyoid-thyroid cartilage shortening rate (ASR) may be significant indicators affecting the prognosis of patients with dysphagia during the pharyngeal phase after stroke. The combined predictive efficacy of these two parameters demonstrated superior diagnostic performance compared to individual parameter analysis. This enhanced predictive capability may be attributed to the pathophysiological consequences of diminished anterior hyoid excursion and reduced ASR, which collectively contribute to three critical swallowing impairments: compromised epiglottic inversion, insufficient laryngeal elevation leading to inadequate airway protection, and incomplete upper esophageal sphincter relaxation. These physiological alterations significantly elevate the risk of both aspiration events and post-swallow pharyngeal residue accumulation (*Molfenter & Steele, 2013*). This phenomenon was particularly evident in patients with unfavorable clinical outcomes, highlighting the significance of hyoid bone kinematics and ASR as robust prognostic markers for impaired swallowing function recovery. In this study, no significant differences were found in the superior displacement amplitude of the hyoid or the maximum displacement amplitude of the hyoid between the two groups at baseline and post-treatment time points (all *p > 0.05*). This phenomenon may be attributed to the limited efficacy of surface electrical stimulation in activating the thyrohyoid muscle, primarily due to its deep anatomical position within the neck compartment. The muscle's relative inaccessibility to transcutaneous stimulation consequently results in insufficient mechanical force generation for effective superior hyoid displacement. In contrast, the geniohyoid muscle's superficial anatomical positioning facilitates effective activation through surface electrical stimulation. This therapeutic modality enhances the recruitment of fast-twitch (type II) muscle fibers, thereby augmenting geniohyoid muscular strength and significantly improving anterior hyoid excursion parameters (*Pearson Jr, Langmore &*

*Zumwalt, 2010*). The significant prolongation of hyoid movement time in the unfavorable prognosis group may be related to decreased muscle coordination and muscle strength. Long-term enteral feeding status may lead to disuse atrophy of the submental muscles, slowing down the movement speed and acceleration of the hyoid (*Huggins, Tuomi & Young, 1999*) and prolonging the entire movement time. This delayed movement pattern may exacerbate the decline in swallowing efficiency and increase the risk of aspiration. Patients in the unfavorable prognosis group may have more severe nervous system damage, leading to more pronounced impairment of hyoid-related muscle function. Specifically, inadequate contractile force generation in the stylohyoid and digastric muscle complex may significantly compromise anterior hyoid excursion amplitude. Furthermore, the development of disuse atrophy in the submental musculature among patients with poor clinical outcomes (*Kim et al., 2021*) potentiates hyolaryngeal movement impairment and extends total swallowing duration.

There are several limitations to this study. Firstly, the sample size is small and the study is single-centered, thus a larger sample size and multi-center collaboration may be necessary in the future to validate the conclusions of this study. Secondly, this study only analyzed the kinematic characteristics of the hyoid bone during dry swallows, which may introduce measurement errors due to difficulties in initiating swallowing (*Miller & Watkin, 1997*). In future studies, the kinematic analysis of hyoid bone movement with different bolus consistencies or liquid volumes will help to untangle new kinematic features related to the prognosis of patients with post-stroke swallowing disorders. Thirdly, this study did not consider potential confounding factors that may affect swallowing function, such as sarcopenia, nutritional status, and hyoid displacement patterns (*Alves et al., 2022*), which should be taken into account in future research.

## CONCLUSIONS

The measurement of anterior displacement amplitude of the hyoid bone and ASR *via* submental B/M-mode ultrasonography may represent significant factors influencing the prognosis of patients with post-stroke pharyngeal swallowing disorders. These measurements demonstrate good predictive validity for the prognosis of such patients. Hence, submental B/M ultrasonography, as a convenient examination technique with high reproducibility, sensitivity, and specificity, possesses notable clinical application value in the diagnosis and treatment efficacy evaluation of patients with post-stroke pharyngeal swallowing disorders. This imaging modality serves as a valuable adjunct to conventional swallowing assessment methodologies, potentially enhancing comprehensive dysphagia management.

### Funding
The authors received no funding for this work.

## Competing Interests

The authors declare there are no competing interests.

## Author Contributions

- Meng Huang analyzed the data, prepared figures and/or tables, authored or reviewed drafts of the article, and approved the final draft.
- Tong Wu conceived and designed the experiments, performed the experiments, prepared figures and/or tables, authored or reviewed drafts of the article, and approved the final draft.

## Human Ethics

The following information was supplied relating to ethical approvals (*i.e.*, approving body and any reference numbers):

The Medical Ethics Committee of the Affiliated Brain Hospital of Nanjing Medical University (2020-KY162-01).

## Data Availability

The raw measurements are available in the Supplementary Files.

## Supplemental Information

Supplemental information for this article can be found online at http://dx.doi.org/10.7717/peerj.20046#supplemental-information.

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
