# Peer review of "Prognostic utility of submental B/M-Mode ultrasonography for swallowing function assessment in post-stroke pharyngeal dysphagia: a preliminary study"

_PeerJ, doi:10.7717/peerj.20046_

## Round 0.1 · original submission · Major Revisions

Reviewer 1 ·

Basic reporting

Regarding the abstract, I have two main points:
i) While I don't think there is the need to specify that not significant findings are p > 0.05, I suggest to report the raw p values (two or three decimals) for the significant findings (< 0.05) as it is commonly requested to avoid reporting only based on the cut-off.
ii) It is not common to report a "list" in the abstract, therefore I recommend to revise the section discussing the results of the ROC analysis according to standard abstract structure

Experimental design

Although the authors express as a limitation the sample size, it should be reported how this sample size was reached, if they provided an "a priori" analysis or estimation, etc.

In the results section, the same comment regarding the reporting of the p values should be considered.

Validity of the findings

No comment

Additional comments

I don't have further comments

Reviewer 2 ·

Basic reporting

Language and Clarity:
The manuscript uses professional English and is largely clear and grammatically sound. Some phrasing could benefit from minor refinement for flow. Sentence structure is mostly unambiguous, though long paragraphs in the Introduction and Discussion can be broken up for readability.
Literature and Context:
Relevant prior literature is well-cited except on certain references which needs to be cross checked. There is a clear presentation of the knowledge gap (like limited prognostic utility data on ultrasonographic parameters).
Structure and Format:
The manuscript follows a logical structure: Abstract, Introduction, Methods, Results, Discussion, and References. Figures and Tables are well-organized, clearly labelled, and relevant. Raw data and methodological details are thoroughly included, fulfilling data transparency requirements.

Experimental design

The research question is meaningful and explicitly stated: determining the prognostic value of specific hyoid movement metrics.
Ethical approval is clearly documented (2020-KY162-01). Inclusion/exclusion criteria are well-defined. Swallowing therapy and ultrasound protocols are detailed and evidence-based with minor additions needed.
Methodology is described well.
Technical aspects of ultrasound imaging, patient positioning, and data extraction using software are well written with minor recommendations on technical aspects.

Validity of the findings

Appropriate use of statistical tests: t-test, Mann-Whitney U, chi-square, logistic regression, and ROC analysis. Power calculation is not discussed—although this is a preliminary study, a comment on this would strengthen the validity assessment.
Robustness of Data:
Tables are comprehensive and support the writings in the text.
Limitation Disclosure:
Discussion section transparently outlines study limitations such as single-center design, dry swallows only, absence of sarcopenia or data on nutrition.

Additional comments

Strengths:
Highly relevant and timely study addressing a practical, non-invasive tool for dysphagia prognosis. Excellent methodological design, especially in ultrasonography section.
Suggestions for Improvement:
Power analysis: Even a post-hoc comment on effect size or required sample size would contextualize the robustness of findings.

Annotated reviews are not available for download in order to protect the identity of reviewers who chose to remain anonymous.

---

## Round 0.2 · accepted · Accept

Thank you for your efforts in addressing the reviewer feedback. Your manuscript is now accepted for publication.

Reviewer 1 ·

Basic reporting

Thank you. I do not have any further comments or requests.

Experimental design

-

Validity of the findings

-

Reviewer 2 ·

Basic reporting

Clear and unambiguous reporting. References have been added and edited wherever suggested.

Experimental design

Experimental design was meticulously planned and carried out. Adequate review of the literature has been done. Ethical standards have been followed as well. Methods have been described and edited accordingly.

Validity of the findings

-